# Observation of a mixed close-packed structure in superionic water

The study of superionic (SI) water has been a highly active research area since its theoretical prediction. Despite significant experimental and computational efforts, its melting curve and the stability of different oxygen lattices remain debated, impacting our understanding of SI ice's peculiar transport properties. Experimental results at lower pressures show disagreement, whereas data at higher pressures are scarce due to the extreme challenges of such experiments. In this work, we present ultrafast X-ray diffraction results of water compressed by multiple shocks to pressures up to ~ 180 GPa. At pressures exceeding 150 GPa and temperatures around 2500 K, our diffraction patterns challenge the pure FCC-SI phase model, providing experimental evidence of the mixed close-packed superionic phase predicted by advanced ab initio calculations. At lower pressures, we observe simultaneous signatures of BCC and FCC structures within a pressure-temperature range consistent with some static-compression experiments, helping to resolve contradictory results in literature. These insights offer new constraints on the stability domains of SI phases and reveal detailed structural features, such as stacking faults. Our results advance the structural understanding of high-pressure SI ice to a level approaching that of ice I polymorphs, with potential implications for water-rich interiors of giant planets.

Under extreme pressures and temperatures, water transforms into a variety of exotic phases, including the superionic (SI) phase, characterised by a crystalline oxygen lattice permeated by highly mobile protons. This phase, with its hybrid solid-liquid properties, has profound implications for fundamental physics and chemistry, material science, and planetary science, particularly in understanding the interiors of ice giants like Uranus and Neptune. However, despite decades of research, the phase diagram of superionic water remains poorly understood, with considerable controversy surrounding competing phases, precise phase boundaries, and transformation mechanisms. As water is the archetype of hydrogen-rich compounds, its phase behaviour has inspired continuous investigation, leading to the discovery of at least 19 (partially) crystalline phases under different thermodynamic conditions[1]. Therefore, addressing uncertainties on the water SI behaviour is critical for a general understanding of such an exotic state.

Early theoretical work suggested a uniform superionic phase inheriting the body-centered cubic (BCC) oxygen lattice of ice VII and X[2–4]. This belief was seriously questioned after thermodynamic integration-based free energy calculations predicted that a face-centred cubic (FCC) oxygen lattice is thermodynamically stable and could potentially affect transport properties[5]. Adding to the complexity, theoretical studies have proposed the existence of competing phases within the SI regime. Ab initio calculations by Sun et al.[6] and Cheng et al.[7] suggest that lower symmetry close-packed structures, such as stacking-disordered phases, may compete with or bridge the pure SI-BCC, SI-FCC, or SI-HCP phases. Characterising these faulted phases is essential, as dislocations can significantly influence viscosity, thus potentially affecting the interior dynamics of ice giants[8]. Interestingly, stacking disorder has also been observed in the water ice Ih and Ic polymorphs, forming ice Isd, which might be encountered in

✉e-mail: leon.andriambariarijaona@polytechnique.edu; michael.stevenson@uni-rostock.de; dominik.kraus@uni-rostock.de; alessandra.ravasio@polytechnique.edu

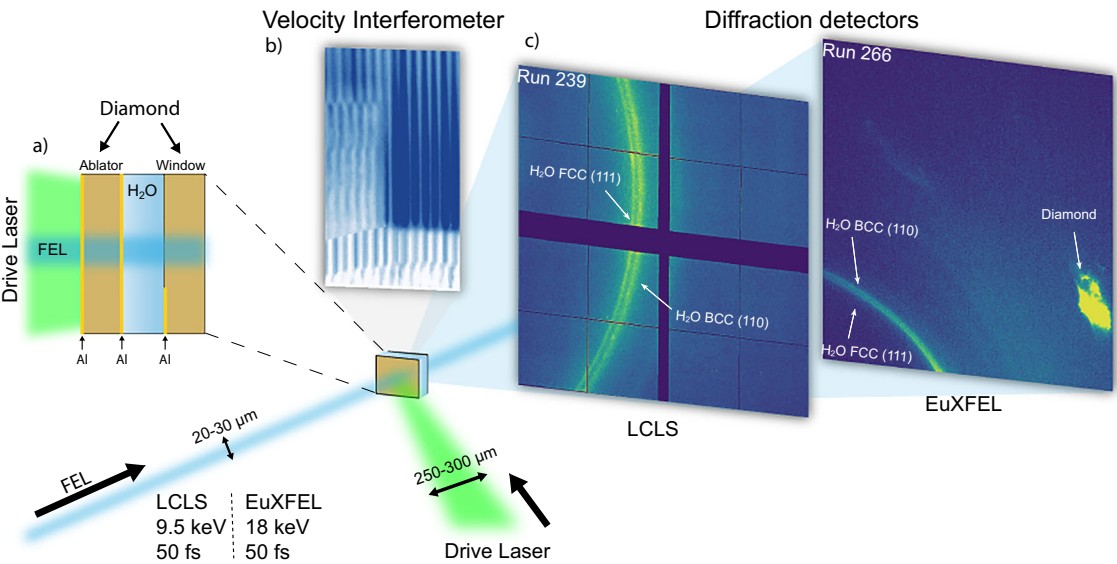

**Fig. 1 | Schematic of the experimental set-up.** Simultaneous shock compression, X-ray diffraction and velocimetry measurements at LCLS and EuXFEL. Water is compressed multiple times by laser-generated shock waves reverberating between diamond windows (**a**). The VISAR data (**b**) track the evolution of the velocity of various interfaces in the target, among which the free surface of the diamond window $u_f(t)$. These measurements are used to infer the pressure-density compression path. The microscopic state of the water sample is probed by a single 50 fs X-ray pulse with a photon energy of 9.5 keV at LCLS, and 18 keV at EuXFEL. X-ray diffraction is recorded using a large-area detector (**c**), and it allows investigating the structure of the compressed water.

various environments in the Solar System, including the outermost ice-rich layer of icy moons[9,10]

Experimentally probing the SI phase at the extreme pressures (>100 GPa) and temperatures (>2000 K) of planetary interiors poses significant challenges and the aforementioned predictions remain unconfirmed. Even at relatively low pressures, where diamond anvil cell (DAC) experiments allow robust phase determination, discrepancies abound. For instance, Prakapenka et al. [11] observed the SI-FCC phase at pressures as low as 30 GPa, while Weck et al. [12] found no evidence of SI-FCC below 50 GPa. The phase boundaries reported in these studies diverge significantly, with Prakapenka et al. observing transition temperatures that are 600 K higher than those reported by Weck et al. at 60 GPa. Most recently, Forestier et al. [13] have characterised the BCC-FCC phase boundary and successfully recovered metastable FCC ice, thereby enabling a precise determination of the superionic transition onset. At higher pressures, the situation becomes even more complicated due to the experimental limitations of static compression methods. Dynamic compression techniques, such as laser-driven shock loading, have extended the exploration of the SI phase to higher pressures, but results remain inconsistent. Millot et al. [14] reported the FCC-SI phase at 160 GPa and 3200 K, whereas Gleason et al. [15], using a similar technique, observed a BCC phase under similar pressures and ~2700 K. The BCC phase would remain stable at 200 GPa and 3200 K up to temperatures exceeding 5000 K. In general, dynamic compression experiments face inherent challenges in achieving high-quality in-situ X-ray diffraction data under extreme conditions, which hindered the ability to unambiguously identify crystalline phases and to resolve microstructural features such as mixed phases or stacking faults.

In this work, we present ultrafast X-ray diffraction measurements of water compressed by reverberating shocks, achieving an unprecedented level of resolution for dynamic compression experiments. By exploring a broad region of the phase diagram within the predicted superionic regime, our data complement and refine interpretations from previous experiments. Our results at $P \geq 150$ (10) GPa and $T \geq 2450$ (135) K reveal a predominantly FCC lattice, consistent with the findings of Millot et al.[14]. However, the observed diffraction patterns deviate from the pure SI-FCC model and require the inclusion of stacking disorder to fully explain the data. This provides experimental evidence of a mixed close-packed phase in the superionic regime, as predicted by advanced ab initio calculations[6,7]. At lower pressures, our results supply critical data for the refinement of the high-pressure phase diagram of water, addressing the complex mechanisms underlying the SI phase transitions.

## Results and discussion

The results reported here were obtained by coupling laser-driven compression with high-brilliance X-ray probes from free-electron lasers (Fig. 1). The thermodynamic conditions of the samples were constrained using optical diagnostics routinely applied in dynamic compression studies and correlated with time-resolved X-ray diffraction measurements (Fig. 2). This experimental configuration allows us to reach extreme pressures and temperatures while simultaneously capturing the structural evolution of the material on sub-nanosecond timescales. Comprehensive details of the experimental setup, diagnostic configuration, and analysis procedures are given in the 'Methods' section and Supplementary Information. Figure 3 presents representative diffraction patterns recorded under different compression conditions. Additional data are reported in the Supplementary Information. The diffracted signal from diamonds is minimised mainly by using single crystal windows instead of polycrystalline samples (see Supplementary Information Fig. S6). Moreover, diamond contributions are readily distinguishable from those originating from the water ice, due to their distinct textures: the diamond features appear as characteristic 'textured', while the water ice produces very 'powder-like' patterns. This distinction is particularly clear in the data shown in the lower panel of Fig. 3, where the feature near 4.2 Å$^{-1}$ cannot be attributed to any diamond reflections but it is instead consistent with the 200 reflection of BCC ice (further details are in the Supplementary Information Section 4.1).

Results for the maximum compression achieved in our campaigns, at P ~180 GPa and temperature of 2500–3000 K, are shown in Fig. 3a. Similar diffraction patterns were observed for pressures exceeding 150 GPa in the same temperature range, recorded across four independent experiments in the two campaigns (see Fig. 4b and Supplementary Information Fig. S12). These data clearly lack peaks

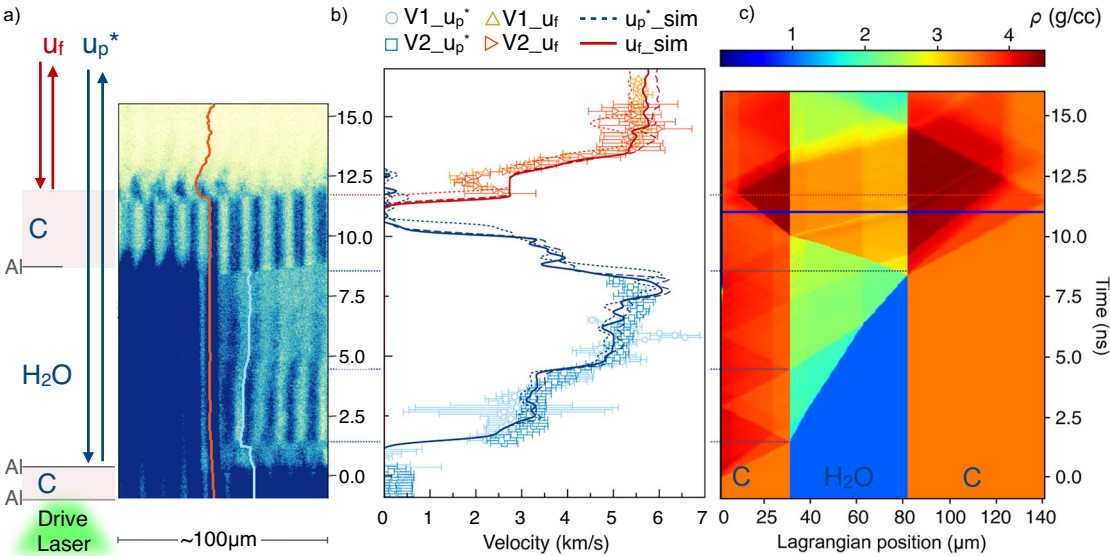

**Fig. 2 | VISAR data and hydrodynamic simulations. a** VISAR image for a target with Al coating of 100 nm added to the surface of the diamond ablator and on half of the rear diamond window surface in contact with water. Fringe shifts from the rear side of the diamond window (red curve) and from the interface at the diamond ablator-water interface (blue curve) are linked to the diamond window free surface ($u_f$) and to the particle velocity ($u_p$), respectively. Fringes between -8 ns and -11.5 ns are ghosts coming from the steady rear side diamond surface before shock breakout. **b** corresponding velocity values from the two VISARs systems VISAR 1 (V1) and VISAR 2 (V2) for apparent velocity $u_p^*$ (open blue circles, V1, and squares, V2) and $u_f$ (orange right- and up- pointing open triangles for V1 and V2) together

with the apparent velocity of the diamond ablator-water interface (blue curve) and rear diamond velocity (red curves) given by the optimised ESTHER hydrodynamic simulation. Uncertainties in the velocity measurements correspond to the standard deviation of fringe-shift values within the analyzed region. Dotted, dashed, and solid lines represent different simulations within VISAR measurement uncertainties. **c** density map of the corresponding hydrodynamic simulation. The blue line corresponds to the probing time for Run 346 at LCLS, shown in Fig. S12d. Other examples can be found in the Supplementary Information, Fig. S1. The *y*-axis (time ns) is common for the tree (**a**–**c**).

associated with the BCC phase, ruling out its stability in this region of the phase diagram[15]. Instead, reflections corresponding to the 111, 200, and 220 planes of the FCC phase are observed, in line with earlier results obtained under comparable conditions[14]. However, the unprecedented spectral resolution of our data reveals that a single FCC phase is insufficient to fully explain the diffraction patterns. Considering contributions from the diamond windows, liquid signal, or grain-size effects do not allow a comprehensive understanding of the data (see Supplementary Information Section 6). Instead, the patterns are best explained by a model combining FCC and lower symmetry structures, such as hexagonal close-packed (HCP). The broadening of the HCP-related peaks relative to those of the FCC phase suggests a highly complex and disordered structure (see Supplementary Information Fig. S10). This observation suggests that the features are better explained by stacking disorder (SD) between FCC- and HCP-like sequences rather than by the coexistence of distinct phases[16–18]. This interpretation is reinforced by the remarkable agreement between our data and the stacking faults (SF) refinement, as illustrated in Figs. 3a and 4b. Further evidence for this scenario is also provided by the strong similarity between our diffraction pattern and machine-learning-potential-based simulations of mixed stacking, shown as yellow and blue curves in Fig. 4. The stacking disorder results from variations in the stacking sequence of the atomic planes, which can alternate locally between structures (see Fig. 4c inset). This behaviour in SI water has been predicted by theoretical simulations, which suggest that within specific pressure and temperature ranges—typically between 100 GPa and 800 GPa, and temperatures from 2000 K to 5500 K—lower symmetry SI stacked structures can compete with the FCC-SI phase[6,7]. In these predictions, the oxygen sublattice may alternate between different stacking configurations, predominantly HCP (AB), FCC (ABC), and dhcp (ABAC), leading to stacking defects within the crystals. Furthermore, in their simulations, Sun et al.[6] report that these different structures are so similar that they can transform from

one configuration to another. As a result, they were unable to distinguish between these structures, referring to a single 'close-packed' (CP-SI) phase. This assumption also holds for our data, and the FCC/HCP stacking fault phases were used to model the data. The stacking disorder is quantified by a factor $\alpha$, representing the probability of observing an HCP sequence among three random layers. A perfect FCC crystal corresponds to $\alpha = 0$ (ABCABC sequence), while a perfect HCP crystal has $\alpha = 1$ (ABABAB sequence)[16,17] (see Fig. 4c). In our data set, $\alpha$ values range from 0.25 to 0.32, indicating a predominant FCC phase with 25–32% HCP contributions (see Supplementary Information section 5 for more details). These results suggest significant local stacking disorder within the lattice. However, the refinement assuming simple stacking faults does not fully capture all the experimental features. Small but systematic discrepancies are observed between the calculated and experimental patterns, notably near Q ~3.4 Å⁻¹ and around the (200)/(110) reflections (see Fig. 4b and Supplementary Information Fig. S12). This suggests that the local structure is more complex than a simple FCC-HCP stacking sequence, and that more sophisticated models may be required to accurately reproduce the diffraction features.

From our results, we cannot discriminate whether the observed stacking disorder arises from the strong uniaxial, high-strain-rate compression associated with shock-driven pressure, as for example observed in copper and gold[19,20], or if it represents an intrinsic feature of superionic ice.

Given that stacking-disordered ice I (referred to as ice Isd) may be very widespread[10], this structural feature could be a fundamental characteristic of water ice, suggesting that even at high pressures its behaviour is far more complex than previously understood. Kinetic effects similar to those observed in ice Ic[21], where disordered cubic and hexagonal stacking sequences gradually lose cubic disorder, transforming over time into the more stable hexagonal phase, may also influence the detection of stacking fault, hindering its observation in

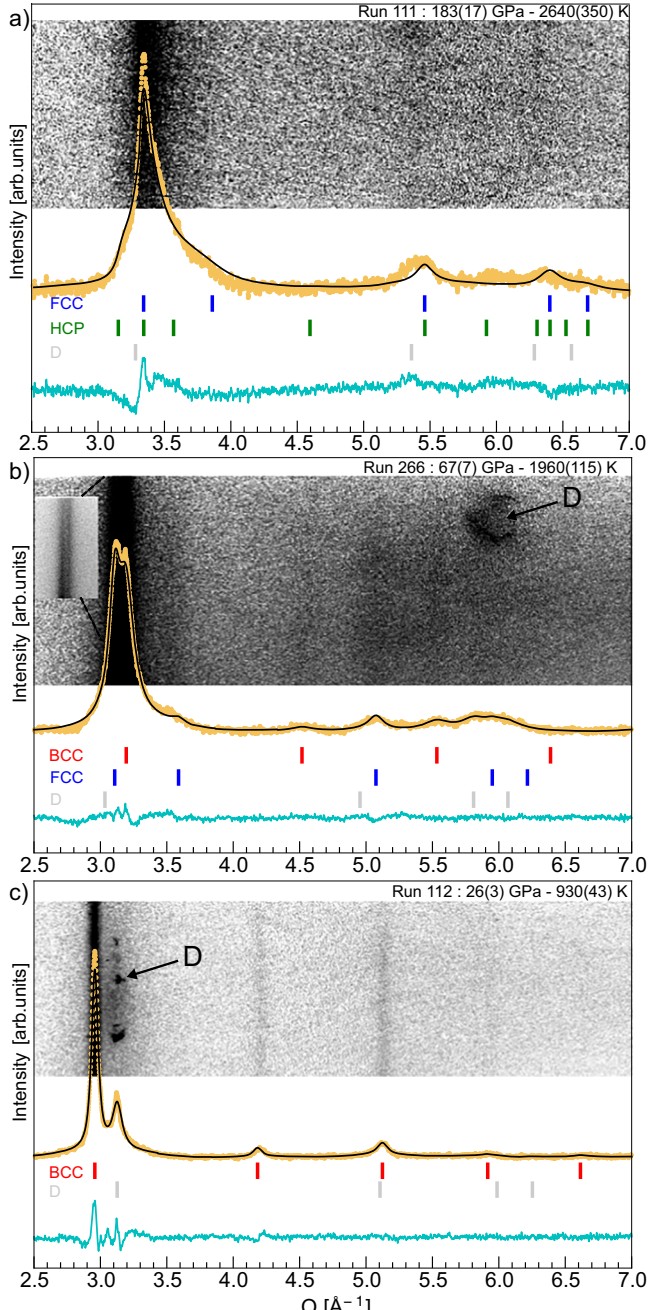

**Fig. 3 | Representative XRD patterns for water at different thermodynamic conditions.** Azimuthally integrated intensities (yellow curves) are overlaid on the $(\phi - Q)$ projections of the experimental data. Continuous black lines correspond to Le Bail refinements in (**b**, **c**), and to a stacking fault (FCC/HCP) refinement in (**a**). Vertical ticks mark the expected Bragg positions for BCC (red), FCC (blue), HCP (green), and the diamond window (grey). In **b**, contrast is enhanced to reveal weaker high-angle peaks; the inset shows the 110 and 111 reflections of the BCC and FCC phases with optimised contrast. In all figures, the cyan line shows the difference between the experimental and calculated intensities. The D indicates contributions from the diamond windows.

static measurements[13]. In this case, the fast temporal scale of our experiment (ns) provides an optimal means to capture such details in the transformation dynamics.

Varying the laser energy and probing time, we can explore other regions of the phase diagram. Between ~70 and 120 GPa and at temperature ranging from 2000 to 2500 K, the diffraction patterns exhibit

a notable change. In addition to the 111, 200 and 220 reflections of the FCC structure, we found characteristic peaks of the BCC structure, as shown in Fig. 3b and the Supplementary Information. Differently from the higher pressure data discussed above, here the width of the BCC and FCC peaks is similar (see Supplementary Information Fig. S8). Remarkably, both structures also exhibit the same density within the measurement uncertainties (see Supplementary Information Table S1). This observation is consistent with previous studies[11,12,22], some of which[12] attributed the coexistence of the two phases to a temperature gradient. However, since the occurrence of similar temperature and pressure gradients seems unlikely in our experiments, this finding raises the possibility that BCC-SI and FCC-SI structures may coexist locally, prompting further discussion on the underlying mechanisms and kinetics. Indeed, in this pressure/temperature range ab initio-based calculations find very similar chemical potentials $\mu$ for the two phases and even $\mu_{bcc} = \mu_{fcc}$ in the hatched area in Fig. 5. This provides a plausible explanation for the simultaneous observation of BCC- and FCC-like signatures in our experimental data within this region of the P-T diagram (Fig. 5). In addition, they predict that the BCC arrangement remains kinetically stabilised, while the FCC structure is thermodynamically favoured[7].

At even lower laser energies and longer delays, our diffraction data provide unequivocal evidence for the presence of a sole BCC structure. As shown in Fig. 3c and Supplementary Information Fig. S7 up to four distinct peaks are clearly observed. VISAR-calibrated hydrodynamic simulations place the experimental conditions between ~25 and 50 GPa and at temperature ranging from 900 to 1300 K. The densities inferred from the XRD (2.153 (9), 2.207(2), and 2.371(8) g/cm³, respectively) at the simulated temperatures are consistent with BCC-SI (see Supplementary Information Fig. S17). These findings are in line with data available in literature[11,12,22,23].

In Fig. 5, our data ensemble is plotted in the P-T diagram alongside previous results. To ensure clarity and effectively convey our findings, we present only a subset of existing experimental and theoretical results. A more comprehensive comparison with additional data is provided in the Supplementary Information (Fig. S18).

Our measurements corroborate the BCC-FCC phase boundary established by Weck et al.[12], Forestier et al.[13] and Millot et al.[14], in disagreement with the higher-temperature transition proposed by Prakapenka et al.[11], Husband et al.[22] and Gleason et al.[15] (Supplementary Information Fig. 18). For pressures and temperatures between 148(10)-180(20) GPa and 2445(150)-2700(350) K we identify a mixed close-packed structure, providing experimental support for the complex phase behaviour predicted by ab initio calculations of Cheng et al.[7] and Sun et al.[6]. The high resolution of our XRD data allows us to revisit previous shock compression results. We unravel HCP stacking disorder within the FCC-SI phase previously identified by Millot et al.[14], while contesting the stability of the BCC-SI in this P-T regime as suggested by Gleason et al.[15]. If intrinsic, the observed stacking disorder would affect the transport properties that govern internal structure, rheology, and magnetic field generation in ice-rich planets[8,24]. If instead shock-induced, its significance would depend on the lifetime of the metastable faulted phase, a question that our work cannot resolve, but that future investigations should address, given the growing recognition of giant impacts as a key hypothesis for shaping planetary interiors[25,26].

Our work thus provides new critical constraints on water's high-density phase behaviour. It demonstrates a high-pressure high-temperature diagram, shaped by mixed structural phases and kinetic effects, that is significantly more intricate than previously established. The complexity of the superionic regime mirrors the rich phase behaviour of solid ice and reinforces how water, a seemingly simple and ubiquitous molecule, continues to reveal exceptional and unexpected physical properties, with potential impacts on astrophysics and planetary science.

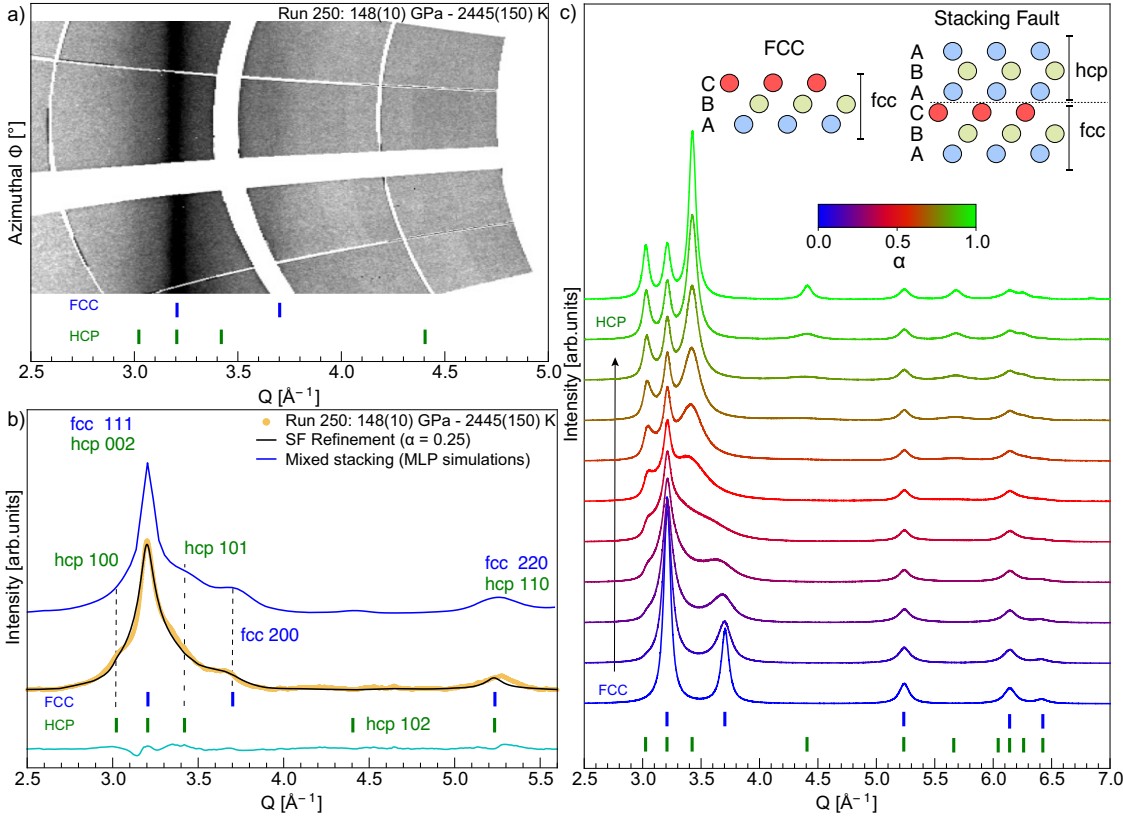

**Fig. 4 | Experimental XRD patterns at high pressure and stacking faults.**
**a**, **b** X-ray diffraction data projected in the $(\phi - Q)$ space, with azimuthally integrated intensities. Experimental data were refined using a stacking fault model incorporating HCP defects with a stacking probability of $\alpha = 0.25$ (black curves). The cyan line shows the difference between the experimental and calculated intensities. The experimental patterns are compared to a simulated XRD pattern

(thick blue line) of a mixed stacking (FCC/HCP) structure calculated with LAMMPS. **c** Simulated XRD patterns as a function of the stacking fault probability $\alpha$, with $\alpha = 0$ for pure FCC and $\alpha = 1$ for pure HCP. In all panels, vertical blue and green lines mark expected Bragg peak positions for FCC and HCP structures, respectively. Inset: Schematic illustration of a simplified stacking fault in an FCC structure, where A, B, and C represent atomic layers.

## Methods

### Shock-compression experiment setup and diagnostics

Two separate experimental campaigns were performed at the Linac Coherent Light Source (LCLS)[27] and at the European X-ray Free-Electron Laser Facility (EuXFEL)[28]. In both campaigns, we used the high-energy nanosecond laser pulses, available at MEC and HED-HiBEF end-stations, respectively, to compress water to the SI regime and probed the atomic structure via ultrafast in-situ X-ray diffraction (XRD) using the XFEL beam. The water layer (30–60 μm thick) was confined between a diamond ablator (30 μm thick) and a diamond window (60 μm thick). A sketch of the target assembly is presented in the inset of Fig. 1a. A shock wave is generated in the diamond ablator by focusing a 10 ns square pulse operating at 527 nm into 250 or 300 μm phase-plate smoothed spots. The peak pressure was varied by changing the total energy delivered to the target, in the range of a few $10^{12}$ W/cm². At these laser intensities elastic and plastic waves are generated in the diamond ablator and reverberate between the diamonds with higher shock impedance. The water layer thus undergoes a multi-step compression. This configuration allowed to probe the regime where the SI phase is predicted to be stable, which is not possible for single shock compression due to the steep rise in temperature.

To obtain the generated pressure and temperature conditions, we used optical data combined with hydrodynamic simulations (see Fig. 2 and Fig. S1 in the Supplementary Information). Two line-imaging Velocity Interferometer System for Any Reflectors (VISARs) were used to record the compression history of each shot, by measuring the shock arrival and exit times in water, the velocity of the diamond

window free surface, and, for some shots, the ablator/water interface velocity. Uncertainties in the VISAR velocity measurements were obtained from the standard deviation of the fringe-shift values within the analysed region. When only timing measurements were used, the uncertainties were derived from the combined errors in timings and sample thickness.

### Hydrodynamic simulations

These measurements were then used to calibrate hydrodynamic simulations and infer the thermodynamic state in the water layer. For each experiment, simulations were run using the experimental laser temporal profile, optimising the intensity and the target thickness within uncertainties to better reproduce measured timings and velocities (see Supplementary Information). The simulated pressure and temperature conditions at the X-ray probing time were considered to be representative of the compressed water layer. The reported error bars on the inferred thermodynamic conditions account for the variability among numerical simulations that yield agreement with the experimental data within the VISAR uncertainties. Simulations were done using the hydrodynamic code ESTHER[29]. For diamond, we used the SESAME 7830 equation of state, implemented with mechanical properties issued from ref. 30. This enables us to account for the strong diamond elastic precursor, crucial for the description of the water compression history. For water, we implemented the recent 'AQUA' equation of state[31]. In the pressure and temperature range of the experiment, 'AQUA' shows a very good agreement with the most established ab initio calculations (see Supplementary Information Fig. S3), which corroborates the simulated conditions in water,

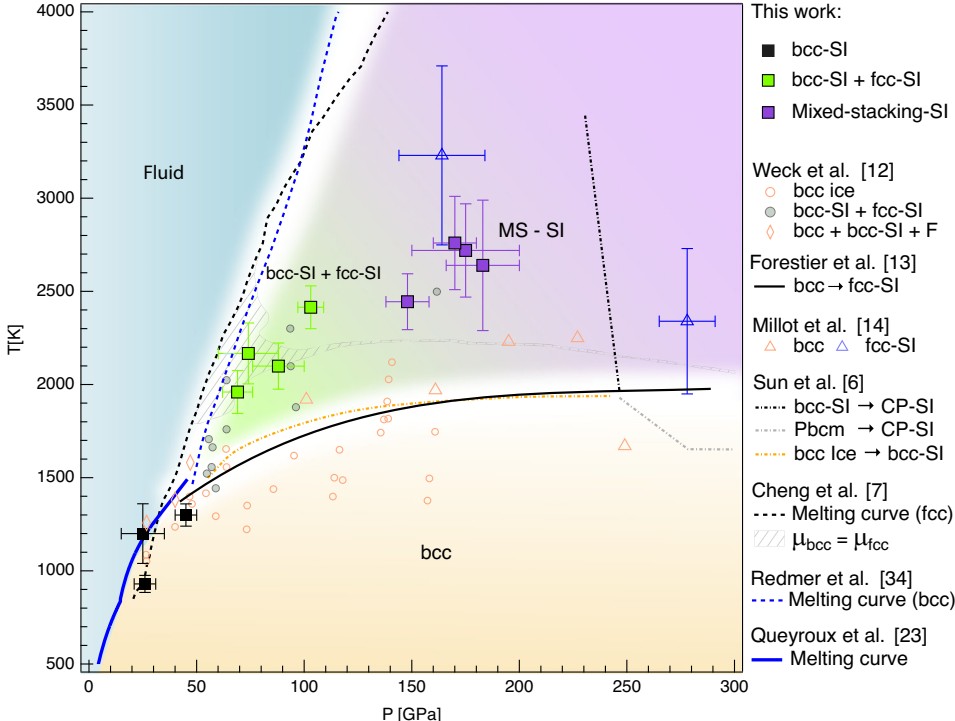

**Fig. 5 | Experimental P-T diagram with the observed H₂O phases.** Square symbols represent data points from this study: black squares indicate the BCC phase, green squares denote regions where BCC and FCC phases coexist, while violet squares indicate areas with observed FCC/HCP. Error bars on P-T conditions represent the variation among simulations consistent with the VISAR uncertainty. Yellow, blue, green, and purple shaded areas are indicative regions for BCC, fluid, BCC+FCC, and Mixed stacking phases, respectively. Triangles show data from dynamic compression experiments: blue and orange triangles correspond to BCC and FCC-SI phases, respectively, as reported in ref. 14. Circles represent results from ref. 12: filled circles indicate coexistence of FCC and BCC phases, while open circles correspond to the BCC phase only. The solid blue and black lines indicate the experimentally determined melting curve and the BCC-FCC transition line, respectively, based on refs. 13,23. Dashed lines represent predicted melting curves from refs. 7,35, and the dash-dotted line shows the theoretical phase boundary proposed by Ref. 6. The hatched area indicates the BCC-FCC coexistence region from ref. 7 (chemical potentials $\mu_{bcc} = \mu_{fcc}$). For additional comparisons with previous experimental and theoretical results, as well as the Pressure-density diagram, see Supplementary Information Figs. S17 and S18.

especially for temperatures, where other widespread EOS, e.g., SESAME tables 7150 or 7154, differ substantially (Supplementary Information Fig. S3).

### X-ray diffraction

In-situ XRD was performed using quasi-monochromatic ($dE/E = 0.2$ −0.5%) 9.5 keV and 18 keV X-ray pulses at the LCLS and EuXFEL, respectively. In both cases, X-ray pulses of ~50 fs duration and ~20−30 μm spot diameter were used, much smaller than the flat-compressed region to minimise pressure and temperature gradients. By varying the timing between the X-ray beam and the drive laser, we probed different stages of the compression history. This approach, combined with different drive laser energies, allowed us to probe a large range of pressure and temperature conditions of the water high-pressure phase diagram.

### Molecular dynamics simulations

Classical molecular dynamics simulations have been performed with the open-source LAMMPS code (stable release version 3 March 2020)[32] interfaced with n2p2[33] to employ a machine learning potential based on density functional theory (DFT) for water. The previously developed water potential was generated by Cheng et al.[7] using a PBE DFT dataset that covers the entire thermodynamic range relevant for this work. Our simulations were carried out for fluid, solid, and superionic phases with BCC, FCC, Pbcm, and mixed structures at various thermodynamic conditions. The number of water molecules considered in the simulation box varied for each studied phase (e.g., up to 13824 molecules in the mixed phase). The simulations were performed in the NpT ensemble, and each calculation was run for 100 ps with a timestep size of 0.25 fs. Every 1000 time steps, a snapshot of all the atomic positions was collected, and the XRD spectrum was calculated for this configuration using the atomic simulation environment (ASE). The final XRD spectrum for each thermodynamic condition was obtained by averaging the 400 snapshots. The simulation input files, i.e., initial structures for each phase and MD parameters, can be found in the repository associated with Cheng et al.[7].

### Data availability

Data recorded for the experiment at the European XFEL will be openly available at https://doi.org/10.22003/XFEL.EU-DATA-004463-00 once the data embargo of the experiment campaign 4463 has been lifted (2026-10-30). Before the end of the data embargo, all relevant raw data will be available from the authors upon request. Source data are provided with this paper and available in the repository https://doi.org/10.6084/m9.figshare.30624377[34]. Additional data available upon reasonable request. Source data are provided with this paper.

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

## Acknowledgements

Use of the Linac Coherent Light Source (LCLS) SLAC National Accelerator Laboratory is supported by the U.S. Department of Energy Office of Science (SC), Basic Energy Science under Contract No. DE-AC02-76SF00515. This research and use of the MEC end station were supported by SC, Fusion Energy Science, under Contract No. DE-AC02-76SF00515. We acknowledge the European XFEL in Schenefeld, Germany, for provision of X-ray free electron laser beam time at the Scientific Instrument HED (High Energy Density Science) and would like to thank the staff for their assistance. The authors are indebted to the Helmholtz International Beamline for Extreme Fields (HIBEF) user consortium for the provision of instrumentation and staff that enabled this experiment. The data are available [10.22003/XFEL.EU-DATA-004463-00]. The authors would also like to thank D. Tondelier (LPICM), S. Ninet (IMPMC), G. Rousse (Collège de France), and P. Talkovski (DESY) for technical support and F. Soubiran for useful discussions. They also acknowledge the access to the Femtosecond laser micromachining at the Institut de Minéralogie de Physique des Matériaux et de Cosmochimie (IMPMC), developed by the 'Cellule Project,' supported by ANR (2010-JCJC-604-01) and the ERC (H2020, Grant No. 724690). This research was supported by the French National Research Agency (ANR) and the Deutsche Forschungsgemeinschaft (DFG) through the projects PROPICE (grant No. ANR-22-CE92-0031 and DFG Project No. 505630685) and the DFG Research Unit FOR 2440. This work was supported by the Helmholtz Association under VH-NG-1141 and ERC-RA-0041. J.L. was supported by GSI Helmholtzzentrum für Schwerionenforschung, Darmstadt, as part of the R&D project SI-URDK2224 with the University of Rostock. The work of S.S. and D.Kr. was supported by Deutsche Forschungsgemeinschaft (DFG) Project No. 495324226. This research was supported by the Centre for Molecular Water Science (CMWS) in an Early Science Project. D. Kr. is funded by the European Union (ERC, MEGACHEM, Grant No. 101171289). Views and opinions expressed are however those of the author(s)only and do not necessarily reflect those of the European Union or the European Research Council. Neither the European Union nor the granting authority can be held responsible for them. The computational work was supported by the French computational centers TGCC and CINES through the GENCI project GEN15731 (Grant No. A0170915731), the North German Supercomputing Alliance (HLRN) under Grant Nos. mvp00018 and mvp00026, and the ITMZ of the University of Rostock. L.B.F., A.E.G., S.H.G., N.J.H., G.G.D., and C.Sc. are supported by US DOE Fusion Energy Sciences under FWP 100182 and 100866. G.D.G. acknowledges support from the DOE NNSA SSGF programme under DE-NA0003960. E.E.M. and A.D. were supported by the UK Research and Innovation Future Leaders Fellowship (Grant No. MR/W008211/1) awarded to E.E.M.

## Author contributions

A.R. and D.Kr. designed the project and the experiments. A.R., D.Kr., M.G.S., D.B., E.C., A.D., G.D., L.B.F., M.Fro., E.G., A.E.G., S.H.G., G.D.G., N.J.H., Z.H., J.-A.H., B.H., O.S.H., D.Kh., H.J.L., J.L., W.L., E.E.M., B.N.,

B.K.O.-O., S.P., C.Sc. A.K.S. and K.V. performed the dynamic compression experiments at LCLS and A.R., D.Kr, M.G.S. M.B., L.L, F.L., T.V., K.A., C.B., A.B.-M., D.B., E.B., T.E.C., A.D., S.D.D.C., M.-L.H., B.H., H.H., O.S.H., R.H., Z.K., J.K., A.L.G., B.L., J.L., M.M., P.M., E.E.M, M.N., J.-P.N., S.P., A.P., T.R.P., C.Q., L.R., D.R., J.R., S.S., J.-P.S., C.St., M.T., T.T., K.V. and U.Z. carried out the experiments at European XFEL. F.L., T.V., Y.G. and K.V. prepared and assembled the targets. L.A. and M.G.S. analysed the XRD data, while T.V. and A.R. analyzed the VISAR data. L.L. and A.R. carried out the hydrodynamic simulations. M.B. and M.Fre. performed the DFT-based molecular dynamics simulations, which were supported by A.B., J.V. and R.R. All authors discussed the results. A.R., L.A., M.G.S., M.B., and D.Kr. wrote the initial manuscript, which was revised and improved by J.-A.H., N.J.H., E.E.M., L.L. and A.D. All authors refined the last version.

## Funding

## Competing interests

The authors declare no competing interests.

## Additional information

L. Andriambariarijaona, M. G. Stevenson, D. Kraus or A. Ravasio.

L. Andriambariarijaona [1,13] ✉, M. G. Stevenson[2,13] ✉, M. Bethkenhagen [1], L. Lecherbourg [3,4], F. Lefèvre[1], T. Vinci [1], K. Appel [5], C. Baehtz [6], A. Benuzzi-Mounaix[1], A. Bergermann [7], D. Bespalov [5], E. Brambrink[5], T. E. Cowan[6], E. Cunningham[7], A. Descamps [8], S. Di Dio Cafiso[6], G. Dyer [7], L. B. Fletcher [7], M. French [2], M. Frost [7], E. Galtier[7], A. E. Gleason [7], S. H. Glenzer [7], G. D. Glenn [7,9], Y. Guarnelli[10], N. J. Hartley [7], Z. He [2], M.-L. Herbert [2,6], J.-A. Hernandez [11], B. Heuser [2,6], H. Höppner [6], O. S. Humphries [5,6], R. Husband [12], D. Khaghani [7], Z. Konôpková[5], J. Kuhlke[2,6], A. Laso Garcia [6], H. J. Lee [7], B. Lindqvist [2,6], J. Lütgert [2], W. Lynn[8], M. Masruri[6], P. May[2], E. E. McBride [8], B. Nagler [7], M. Nakatsutsumi [5], J.-P. Naedler[2], B. K. Ofori-Okai [7], S. Pandolfi[10], A. Pelka[6], T. R. Preston [5], C. Qu[2], L. Randolph [5], D. Ranjan[2,6,12], R. Redmer [2], J. Rips[2], C. Schoenwaelder [7], S. Schumacher [2], A. K. Schuster [6], J.-P. Schwinkendorf[5,6], C. Strohm[12], M. Tang[5,12], T. Toncian [6], K. Voigt[6], J. Vorberger [6], U. Zastrau [5], D. Kraus [2,6] ✉ & A. Ravasio [1] ✉

¹Laboratoire LULI, CNRS - École Polytechnique - CEA - Sorbonne Université, Palaiseau, France. ²Institut für Physik, Universität Rostock, Rostock, Germany. ³CEA, DAM, DIF, Arpajon, France. ⁴Laboratoire Matière en Conditions Extrêmes, Université Paris-Saclay, CEA, Bruyères-le-Châtel, France. ⁵European XFEL, Schenefeld, Germany. ⁶Helmholtz-Zentrum Dresden-Rossendorf, Dresden, Germany. ⁷SLAC National Accelerator Laboratory, Menlo Park, CA, USA. ⁸School of Mathematics and Physics, Queen's University Belfast, Belfast, UK. ⁹Department of Applied Physics, Stanford University, Stanford, CA, USA. ¹⁰Institut de Minéralogie, de Physique des Matériaux et de Cosmochimie (IMPMC), Sorbonne Université, MNHN, CNRS UMR 7590, Paris, France. ¹¹European Synchrotron Radiation Facility (ESRF), Grenoble, France. ¹²Deutsches Elektronen-Synchrotron DESY, Hamburg, Germany. ¹³These authors contributed equally: L. Andriambariarijaona, M. G. Stevenson. ✉e-mail: leon.andriambariarijaona@polytechnique.edu; michael.stevenson@uni-rostock.de; dominik.kraus@uni-rostock.de; alessandra.ravasio@polytechnique.edu

