## [Transparent Peer Review file · Nature Communications]

Observation of a mixed close-packed structure in superionic water

Corresponding Author: Dr Leon Andriambarijaona

Version 0:

Reviewer comments:

Reviewer #1

(Remarks to the Author)

Review Comments

The target of this study, superionic ice, has recently been the subject of intense research competition, owing to its dual nature—possessing characteristics of both solids and liquids—as well as its significance in planetary science. The notable strength of this work lies in the use of XFEL and VISAR, which has enabled an unprecedentedly precise determination of temperature and pressure conditions, along with the acquisition of high-quality X-ray diffraction data. Remarkably, under such extreme high-temperature and high-pressure conditions, diffraction patterns are typically “spotty,” yet in this study, the authors obtained homogeneous and ideal “smooth” powder patterns. This has been crucial for the high-precision analysis of ice with stacking disorder, and for the first time, has allowed a truly crystallographic discussion of superionic ice.

I am not a specialist in dynamic compression, and thus I am not in a position to evaluate the validity of the techniques employed in this work, such as VISAR. If the temperature and pressure in this study are indeed determined with higher accuracy and reliability than previous estimates, I have no doubt that the paper will eventually merit publication in Nature Communications. I therefore leave the judgment on this matter to other reviewers with expertise in dynamic compression. That said, there remain several issues of concern that should be carefully addressed prior to publication.

Interpretation of the phase diagram

The primary concern lies in the authors’ claim that FCC and HCP phases coexist as stable phases in the phase diagram. From the viewpoint of classical thermodynamics, as is evident from the Gibbs phase rule ($F - C + P = 2$), for a single-component system ($C = 1$), coexistence of two phases ($P = 2$) leaves one degree of freedom ($F = 1$), meaning that such coexistence is typically allowed only along a phase boundary line in the phase diagram. However, in Fig. 5, a certain P–T region is depicted as a coexistence region of bcc and fcc phases.

Indeed, as the authors argue, under such high-temperature conditions the free energy difference between bcc and fcc phases may be extremely small, and non-equilibrium coexistence of the two phases may occasionally occur. Nevertheless, only thermodynamically stable phases can have their own stability fields in a phase diagram, and it is inconsistent with classical thermodynamics for two phases of different symmetry to both be stable within the same P–T region.

The only possible exception (or out of the classical thermodynamic) would be if the bcc and fcc structures were intermixed on a scale smaller than ~100 nm, with no long-range periodicity maintained—essentially a nanostructured mixture. Such a nanoscale “phase separation” would not constitute a macroscopically homogeneous thermodynamic phase, and should perhaps be regarded differently in the context of a phase diagram. How to treat such nanoscopic phase-separated states in phase diagrams may be an open question for future condensed matter physics.

A similar argument applies to the mixed-stacking SI phase. At such high temperatures, it is entirely plausible that stacking disorder could be energetically more favorable than either pure fcc or hcp phases, due to the entropic gain. Stacking disorder, by its nature, lacks long-range periodicity along a given axis and therefore has no macroscopic symmetry; changes in the stacking probability α do not alter the overall symmetry. Consequently, one might interpret it as a single homogeneous phase. Indeed, in the case of ice I_{sd}, Lupi et al. (2018, Nature 551, 241–245; doi:10.1038/nature24279) have pointed out that ice I_{sd} could be more stable than either pure I_h or I_c under certain temperatures and domain sizes.

In light of the above, the discussion of Fig. 5 and the purported coexistence of bcc and fcc phases should be reconsidered.

Minor points

- The manuscript discusses possible implications for “giant impacts.” Could the authors support this statement with examples—preferably from the metallurgy literature—where stacking disorder affects mechanical properties? Relevant citations would be welcome. Furthermore, given that ice produced under shock compression is likely to revert rapidly to ambient-pressure phases, do its mechanical or transport properties still hold planetary significance during its brief lifetime? Overly speculative implications may risk diminishing the paper’s value and should be carefully considered.
- In Fig. S5, the CeO₂ profile does not fit well. Both the high-Q and low-Q regions appear to deviate in peak positions (with asymmetric residuals at high-Q). This may be improved by optimizing the peak shift parameters and the profile function. Although fixing $U = 0$ seems reasonable given the limited 2θ range, V should likely be treated as a variable. Also, in the caption to Fig. S5, “CeCO₂” should be corrected to “CeO₂,” and X and Y may not be asymmetric parameters but Lorentzian terms; please confirm.
- Throughout the manuscript, reflection indices, hkl , should not be enclosed in parentheses (see the introduction to International Tables for Crystallography, Vol. A).
- The terms “stacking fault(s)” and “stacking disorder” should be used distinctly. “Stacking fault(s)” refers to individual defect(s) on one or plural layer(s) with differently aligned to surrounding layers, whereas “stacking disorder” refers to the state of a material containing many such faults, may be defined by detectable using diffraction methods. In most cases in this manuscript, “stacking disorder” would be the more accurate term rather than “stacking fault(s)”.
- P-T conditions may be included in the legend in respective figures as well as run numbers.
- Several figure references appear as “Fig. ??.” Please check and correct them.
- On page 2 of the Supplementary Materials, “ice VII/VII* X ” is written. What does VII* mean? (Is it a typographical error for VII’?) Since the manuscript does not explain VII’, a definition should be provided.

Reviewer #2

(Remarks to the Author)
please see attachment

Version 1:

Reviewer comments:

Reviewer #1

(Remarks to the Author)

I confirm that all the concerns I raised in the previous review have been appropriately corrected. In addition, it appears that the comments from other experts on the dynamic compression experiments have been adequately addressed. There are no further points raised by me and I can conclude that the manuscript can be ready to be published.

Reviewer #2

(Remarks to the Author)

The authors have provided detailed responses to the questions I had on the original manuscript. I thought the additional analysis they report in their response to my comments: showing that the FWHM of HCP is on average broader than FCC is interesting and I thought they might consider adding this to their SM? Beyond this, I'm happy to recommend publication.

Response to Reviewers : manuscript NCOMMS-25-52819-T

We would like to begin by sincerely thanking the referees for their thoughtful comments and detailed suggestions, which have been very helpful in further refining our diffraction analysis. We are also grateful for their positive assessment of our study. Acquiring diffraction data of this quality under shock-compression conditions is far from routine, and it is especially encouraging that expert referees have recognized the significance of our results. Their observations that “*it is impressive that it is possible to extract data of this quality from such extreme conditions*” and that this has “*for the first time, allowed a truly crystallographic discussion of superionic ice*” are both motivating and validating for our work.

In the following, we provide detailed point-by-point responses to all referees’ comments and questions (our responses appear in blue). For clarity, all significant changes made in the revised manuscript are also highlighted in blue.

Response to the reviewers

Reviewer 1

The target of this study, superionic ice, has recently been the subject of intense research competition, owing to its dual nature—possessing characteristics of both solids and liquids—as well as its significance in planetary science. The notable strength of this work lies in the use of XFEL and VISAR, which has enabled an unprecedentedly precise determination of temperature and pressure conditions, along with the acquisition of high-quality X-ray diffraction data. Remarkably, under such extreme high-temperature and high-pressure conditions, diffraction patterns are typically “spotty,” yet in this study, the authors obtained homogeneous and ideal “smooth” powder patterns. This has been crucial for the high-precision analysis of ice with stacking disorder, and for the first time, has allowed a truly crystallographic discussion of superionic ice. I am not a specialist in dynamic compression, and thus I am not in a position to evaluate the validity of the techniques employed in this work, such as VISAR. If the temperature and pressure in this study are indeed determined with higher accuracy and reliability than previous estimates, I have no doubt that the paper will eventually merit publication in Nature Communications. I therefore leave the judgment on this matter to other reviewers with expertise in dynamic compression. That said, there remain several issues of concern that should be carefully addressed prior to publication.

Reviewer Comment 1.1 — Interpretation of the phase diagram The primary concern lies in the authors’ claim that FCC and HCP (BCC) phases coexist as stable phases in the phase diagram. From the viewpoint of classical thermodynamics, as is evident from the Gibbs phase rule ($F - C + P = 2$), for a single-component system ($C = 1$), coexistence of two phases ($P = 2$) leaves one degree of freedom ($F = 1$), meaning that such coexistence is typically allowed only along a phase boundary line in the phase diagram. However, in Fig. 5, a certain P–T region is depicted as a coexistence region of bcc and fcc phases.

Indeed, as the authors argue, under such high-temperature conditions the free energy difference between bcc and fcc phases may be extremely small, and non-equilibrium coexistence of the two phases may occasionally occur. Nevertheless, only thermodynamically stable phases can have their

own stability fields in a phase diagram, and it is inconsistent with classical thermodynamics for two phases of different symmetry to both be stable within the same P-T region.

The only possible exception (or out of the classical thermodynamic) would be if the bcc and fcc structures were intermixed on a scale smaller than 100 nm, with no long-range periodicity maintained—essentially a nanostructured mixture. Such a nanoscale “phase separation” would not constitute a macroscopically homogeneous thermodynamic phase, and should perhaps be regarded differently in the context of a phase diagram. How to treat such nanoscopic phase-separated states in phase diagrams may be an open question for future condensed matter physics.

A similar argument applies to the mixed-stacking SI phase. At such high temperatures, it is entirely plausible that stacking disorder could be energetically more favorable than either pure fcc or hcp phases, due to the entropic gain. Stacking disorder, by its nature, lacks long-range periodicity along a given axis and therefore has no macroscopic symmetry; changes in the stacking probability α do not alter the overall symmetry. Consequently, one might interpret it as a single homogeneous phase. Indeed, in the case of ice Isd, Lupi et al. (2018, *Nature* 551, 241–245; doi:10.1038/nature24279) have pointed out that ice Isd could be more stable than either pure Ih or Ic under certain temperatures and domain sizes. In light of the above, the discussion of Fig. 5 and the purported coexistence of bcc and fcc phases should be reconsidered.

Reply:

We are very grateful to the Reviewer for raising this important point.

In our shock experiments, we observe coexistence of the bcc and fcc structures in a pressure/temperature regime, where, according to theoretical predictions, the free-energy difference between the two phases becomes extremely small. We therefore interpret this domain not as a stability field in the thermodynamic sense, but as a region of metastable or non-equilibrium coexistence reflecting the structures experimentally detected. Metastability appears to be a common feature of high-pressure ice: in static compression studies, for example, the fcc phase is known to persist metastably over the bcc phase at room temperature, despite not being the lowest free-energy phase (**Forestier et al. PRL. 2025**)

We fully agree with the Reviewer that, in light of this discussion, Figure 5 should not be regarded as a phase diagram in the strict thermodynamic sense, since we cannot determine the equilibrium stability of the indicated phases. Rather, it should be viewed as a P–T diagram (experimental observation map) showing the structures identified in our measurements. This approach is standard in high-pressure research, including studies using static compression in diamond anvil cells.

To clarify this point, we have revised both the caption of the figure and the main text to explicitly state that Figure 5 represents a P-T diagram (experimental observation map), rather than a thermodynamic equilibrium phase diagram, and that the “region of ”BCC + FCC” corresponds to experimentally observed, likely metastable, coexistence.

Reviewer Comment 1.2 — The manuscript discusses possible implications for “giant impacts”. Could the authors support this statement with examples—preferably from the metallurgy literature—where stacking disorder affects mechanical properties? Relevant citations would be welcome. Furthermore, given that ice produced under shock compression is likely to revert rapidly to ambient-pressure phases, do its mechanical or transport properties still hold planetary significance during its brief lifetime? Overly speculative implications may risk diminishing the paper’s value and should be carefully considered.

Reply: As emphasized on p. 8 of the manuscript, our experiments clearly reveal the presence of stacking disorder. However, the available data do not allow us to determine whether this disorder is

an intrinsic feature of the superionic phase or instead arises from the strong uniaxial, high-strain-rate compression associated with shock loading. Both scenarios are interesting from a fundamental point of view and potentially relevant for planetary science. If the disorder is intrinsic to superionic ice, its consequences for giant planets would be directly linked to the physical properties of the faulted ice phase. In metallurgy, stacking disorder is well known to influence strength and transport properties of metals and alloys (e.g., Su et al., *Mater. Sci. Eng. A* 803, 140696 (2021)). In planetary science, it is well known that crystalline defects play a central role in controlling rheological properties (e.g., Karato et al., *Prog. Earth Planet. Sci.* 12, 27 (2025); Sumita & Bergman, in *Treatise on Geophysics*, Elsevier, 2007) and simulations at extreme conditions are striving to include them (e.g., Li & Scandolo, *Phys. Rev. B* 109, 184108 (2024)). Stacking disorder is also known to affect viscosity and thermal conductivity, including in ice at low pressures (e.g., F. Matusalem et al, *PNAS* 119, e2203397119 (2022), Johari & Andersson, *J. Chem. Phys.* 143, 054505 (2015)). If intrinsic, such effects could therefore directly impact models of planetary interiors, transport properties, and even dynamo generation. Alternatively, if the disorder is shock-induced, it could provide important insight into the physics of planetary collisions, which cannot be reproduced by static compression alone. For instance, shocked feldspars or silicates in meteorites form maskelynite or diaplectic glass (e.g., Rubin, *Icarus* 257, 221–229 (2015)), while static experiments yield only crystalline phases. Similarly, iron alloys form equilibrium crystalline structures under static high-pressure experiments but can adopt metastable phases under shock (e.g., Boettger & Wallace, *Phys. Rev. B* 55, 5 (1997)). By analogy, shock-induced stacking disorder in superionic ice could represent a transient but relevant state during giant impacts. Whether such a metastable phase persists long enough to influence impact dynamics depends on its lifetime relative to hydrodynamic timescales, a point our current experiments cannot resolve. Given the increasing recognition of giant impacts as a key hypothesis for shaping planetary interiors (e.g., Helled et al., *Space Sci. Rev.* 216, 38 (2020); Liu et al., *Nature* 572, 355–357 (2019); Müller et al., *A&A* 638, A124 (2020); Esteves et al., *Icarus* 429, 116428 (2025); Kegerreis et al., *Astrophys. J.* 861, 52 (2018); Reinhardt et al., *MNRAS* 492, 5336–5353 (2020)), we believe it important to highlight this possibility. We note that longer-duration shock studies at these extreme conditions could be challenging, so that theoretical work and numerical simulations will be essential to address this aspect.

We agree with the referee that this aspect was not adequately discussed in the first version of the manuscript, and we have now changed the sentence at old line 197:

The identified stacking faults may significantly influence the mechanical and transport properties of superionic water [8], particularly under the extreme conditions associated with giant impacts. As a result, our understanding of ice-rich planetary interiors—such as those expected in Uranus and Neptune—could be substantially revised, given their dependence on properties like viscosity and thermal conductivity.

with

*If intrinsic, the observed stacking disorder would affect the transport properties that govern internal structure, rheology and magnetic field generation in ice-rich planets [8, Johari & Andersson, *J. Chem. Phys.* 143, 054505 (2015)]. If instead shock-induced, its significance would depend on the lifetime of the metastable faulted phase, a question that our work cannot resolve, but that future investigations should address, given the growing recognition of giant impacts as a key hypothesis for shaping planetary interiors (e.g., Helled et al., *Space Sci. Rev.* 216, 38 (2020); Liu et al., *Nature* 572, 355–357 (2019)).*

Reviewer Comment 1.3 — In Fig. S5, the CeO₂ profile does not fit well. Both the high-Q and low-Q regions appear to deviate in peak positions (with asymmetric residuals at high-Q). This may be improved by optimizing the peak shift parameters and the profile function. Although fixing $U = 0$ seems reasonable given the limited 2θ range, V should likely be treated as a variable. Also, in the caption to Fig. S5, “CeCO₂” should be corrected to “CeO₂,” and X and Y may not be asymmetric parameters but Lorentzian terms; please confirm.

Reply: We thank the Reviewer for pointing this out. We have repeated the refinement of the CeO₂ calibrant profile, this time keeping U fixed at 0 but allowing V and W to vary, together with the Zero 2θ shift. The rest of the refinement procedure remained unchanged.

This adjustment significantly improved the agreement between observed and calculated profiles across the entire Q-range, including both low-Q and high-Q regions. The final refined profile parameters are: $U = 0.0$, $V = 0.012011$, $W = 0.004258$, $X = 0.058403$, $Y = 0.057187$, $Z_{\text{zero}} = -0.0026$

The fit is now much better than in the previous version, and the asymmetric residuals at high-Q are largely suppressed. We also corrected the caption typo (“CeCO₂” → “CeO₂”) as suggested. Finally, we confirm that in the Thompson–Cox–Hastings pseudo-Voigt function used in FullProf, X and Y are Lorentzian terms (size and strain broadening, respectively), not asymmetry parameters. This new calibration does not affect the main findings of our work. The main changes concern lattice parameters, grain sizes and micro strains, but such adjustments lie within the error bars.

Figure 1: Refinement of the CeO₂ calibrant profile using the Thompson–Cox–Hastings function. Parameters refined were V , W , X , Y , and the Zero 2θ shift, with U fixed to 0. The resulting values are $V = 0.0120$, $W = 0.0043$, $X = 0.058$, $Y = 0.057$, $Z_{\text{zero}} = -0.0026$. The fit quality is significantly improved compared to the previous versio

Reviewer Comment 1.4 — Throughout the manuscript, reflection indices, hkl , should not be enclosed in parentheses (see the introduction to International Tables for Crystallography, Vol. A).

Reply: In the revised manuscript, all reflection indices hkl have been reformatted to follow the convention of the International Tables for Crystallography, Vol. A, i.e., without parentheses.

Reviewer Comment 1.5 — The terms “stacking fault(s)” and “stacking disorder” should be used distinctly. “Stacking fault(s)” refers to individual defect(s) on one or plural layer(s) with differently

aligned to surrounding layers, whereas “stacking disorder” refers to the state of a material containing many such faults, may be defined by detectable using diffraction methods . In most cases in this manuscript, “stacking disorder” would be the more accurate term rather than “stacking fault(s)”

Reply: We thank the referee for this important remark. In the revised version of the manuscript, we now distinguish carefully between stacking fault(s), which refer to local defects in the stacking sequence, and stacking disorder, which describes the overall disordered state of the structure as observed in diffraction. Since our experimental data reveal mainly average structural effects rather than isolated planar defects, we agree that ‘stacking disorder’ is the most appropriate term in most contexts. We have therefore systematically revised the manuscript to adopt this terminology.

Reviewer Comment 1.6 — P-T conditions may be included in the legend in respective figures as well as run numbers

Reply: In the main text figures, the P–T conditions and run numbers are already included. We have updated the supplementary figures accordingly, to ensure that the P–T conditions and run numbers are systematically visible.

Reviewer Comment 1.7 — Several figure references appear as “Fig. ??.” Please check and correct them

Reply: The occurrences of “**Fig. ??**” were due to missing or broken cross-references in the manuscript. We have checked carefully all figure references and correct them to ensure consistency throughout the text.

Reviewer Comment 1.8 — On page 2 of the Supplementary Materials, “ice VII/VII*/X” is written. What does VII* mean? (Is it a typographical error for VII' ?) Since the manuscript does not explain VII', a definition should be provided

Reply: The notation VII* was indeed a typographical error, and it referred to ice VII', as defined in the articles of Hernandez and Caracas, *Phys. Rev. Lett.* (2016) and Queyroux et al. *Phys. Rev. Lett.* (2020). These papers are now cited as a reference for such phase.

Reviewer 2

This paper describes a series of x-ray diffraction measurements of ice subjected to extremes of temperature and pressure via shock compression techniques. The approach is to confine the samples using diamond windows and then subject them to a shock wave triggered by a 10ns laser pulse with a very high energy density. This creates a rather complex environment described as “elastic and plastic waves [reverberating] between the diamonds” . While the sample responds to this, it is probed using a single 50fs pulse from a free electron laser (generated by either the LCLS or XFEL facilities in two separate experiments). Thus, it effectively takes a rapid snapshot, capturing the instantaneous state of the sample. By controlling the exact delay between shock and x-ray, the experimental team thus aim to select the P/T conditions of the measurement (although determining these relies on some ancillary hydrodynamic simulations).

Reviewer Comment 2.1 — One concern I had in interpreting the diffraction data is that they correspond to signal from a gauge volume that is has a 20-30um diameter and thickness equal to that of the sample. Correspondingly, although radial gradients are minimized by the small beam diameter, any axial gradient will be averaged. This seems like an important point, but I did not see this explicitly addressed in the manuscript. If I correctly read the right-hand panel of Fig 2, it looks like there is measurable step in density around 2/3 of the way through the sample of around 5%. Which, if I look at the equations of state in Fig S17, would map on to a 10-15 GPa pressure differential. The authors state that “temperature and pressure gradients seem unlikely in [their] experiments” , but then in the SI it is mentioned that “where the sample thickness was significant, a contribution from the liquid phase was observed” . This surely indicates that there are significant gradients within the sample if liquid and solid phases co-exist. How do the authors reconcile this?

Reply: We thank the referee for raising this important point. Indeed, as correctly noted, the small X-ray beam diameter minimizes radial gradients, while the averaging across the sample thickness may include contributions from any axial gradients. This issue is discussed in Section 3 of the Supplementary Material, but we are happy to provide further clarification here. The presence and importance of axial gradients depend strongly on the hydrodynamic history of the shock compression and the probing time. To reach a representative description of these effects, we combined hydrodynamic simulations with optical diagnostics (see Section 2 of the Supplementary Material, Fig. 2 and Fig. S1). As the referee observes, Fig. 2 does display a density “step”. This feature arises in the LCLS data because reverberations from the 30 μm diamond ablator overtake the first shock waves within the water layer (Fig.S1 a,b,c). As a result, a steep gradient forms, with a colder region produced by more numerous reverberations, and a hotter region resulting from fewer compressions. In this case, two sets of distinct thermodynamic conditions are generated (Fig. S4a,b). Although both regions contribute to the diffraction patterns, we find that the hotter part is close to (or above) melting (Fig. S4a,b and Fig. S18) and primarily adds a liquid background signal. By contrast, the crystalline diffraction signal originates predominantly from the colder region. This interpretation is consistent with the need to include a substantial liquid background component when refining the LCLS diffraction data (Section 4 of the Supplementary Material). In contrast, for the EuXFEL experiments, we used thinner water layers ($\sim 25\text{--}30 \mu\text{m}$), which prevented the overtaking of shock reverberations and produced much more uniform conditions (Fig. S1d and Fig. S4c,d). The resulting diffraction data are consistent with those from the “cold” part of the LCLS experiments and show similar features under comparable conditions (see Fig. S8 and Fig. S9). This provides further support for our interpretation.

Finally, regarding the quoted sentence “temperature and pressure gradients seem unlikely in [their] experiments”: we agree that, when read in isolation, this wording may appear ambiguous. In its context (line 169 of the old version of the manuscript), the full sentence reads: “However, since the occurrence of similar temperature and pressure gradients seems unlikely in our experiments, this finding raises the possibility of a genuine coexistence of BCC-SI and FCC-SI phases, prompting further discussion on the underlying mechanisms and kinetics.” This sentence was specifically included to address the simultaneous observation of BCC and FCC structures in both static and dynamic experiments. Our intention was not to deny the presence of gradients in the shock-compression data, but rather to emphasize that such gradients are unlikely to match those in static experiments.

Reviewer Comment 2.2 — Of course, the data themselves contain information on any strain gradient. The authors helpfully included data and refinement details from a CeO₂ standard, so the instrument resolution can be accounted for. With this information, substantial sample broadening

is clearly present in all the datasets shown in Fig 3, although it is difficult to quantitatively assess this just by looking at the figures and any strain could also be convolved with particle size effects.

Reply: The broadening of the lines associated to the grain size effects has been evaluated and discussed in the supplementary material, section 6.5. We find that the grain size broadening has a negligible effect on the widths of the collected diffraction peaks.

Reviewer Comment 2.3 — The diffraction profiles observed in the higher P/T region of the phase diagram do convincingly show both FCC and HCP features. And it seems reasonable to interpret these in terms of stacking faults. But I have some concerns regarding the quality of the Le Bail fits. All the patterns in Fig 3 seem to contain Bragg features that are much sharper than simulated peaks. These are easily discernable in the difference curves, circled in the image below:

These sharp features appear more-or-less exactly at the expected positions for the respective crystalline phases, so could be explained by a handful of fault-free crystallites, or extended fault-free domains within crystallites with correspondingly sharper peaks. Having a distribution of domain sizes seems to be rather likely.

Reply: We appreciate the referee's detailed comments and for highlighting the differences observed in the curves of Fig. 3. We agree that it is indeed plausible that our data reflect a distribution of domain sizes. However, we believe it is difficult to infer the origin of such a distribution solely from the available data. As the referee points out, similarly sharp features are also visible in the BCC phase (Fig. 3c), where the data can be explained without invoking faults. We therefore consider it more likely that these features arise from the nucleation process associated with compression. Experimentally, sharper features than those obtained from calculated refinements have also been reported for shocked SiO_2 (see Supplementary Fig. 2. of **A.E. Gleason et.al., Nat. Com. 2015**), which may support our interpretation. Nevertheless, we fully agree with the referee that future studies should examine these features in greater detail.

Reviewer Comment 2.4 — There are also significant discrepancies between fit and data in the SF refinements. This is most notable in Fig 4b below. There are clearly “lumps” in the data that are not fitted by the model in the region from 2.7 to 3.5 \AA^{-1} . I think this is probably indicating that the peak profile model is not adequate to capture the complexities of the real system. It seems plausible that different directions in the lattice could retain coherence over longer distances than others.

Reply: We fully agree with the referee that the local structure may exhibit significant complexity that our simple intrinsic stacking model does not capture. As we explicitly state in Section 5 of the Supplementary Material, “the actual microstructure is likely more complex than a simple FCC–HCP layer alternation.” Nonetheless, we chose not to overinterpret our data by attempting to construct a more complex structural model.

Reviewer Comment 2.5 — Usually, the profile parameters in refinement are parameterized in a way that would not capture hkl-dependent effects on peak broadening. I wondered if simple individual peak fits might describe the data better. And these might contain interesting information, for example, I was curious if either the HCP or FCC component had measurably and consistently sharper peaks than the other.

Reply: Thank you for this helpful suggestion. We performed individual pseudo-Voigt fits in Q -space for representative reflections (HCP: 100, 002, 101; FCC: 111, 200), as summarized in Table 1. On average, the HCP reflections are broader (mean $\text{FWHM}_Q \sim 0.27 \text{ \AA}^{-1}$) compared to the FCC reflections (mean $\text{FWHM}_Q \sim 0.10 \text{ \AA}^{-1}$). However, the broadening within HCP is not uniform: HCP(100) is comparably sharp to FCC(111), whereas HCP(002) and HCP(101) are substantially broader. We also observed that the HCP peaks tend to have a shape factor $\eta \approx 0.5$, indicating a more Gaussian-like profile, whereas the FCC peaks are more Lorentzian ($\eta \approx 1$). These results are consistent with what we already presented in the Supplementary Materilas (see SM section 6.2), where separate FCC and HCP fits revealed similar trends, in line with disorder-dominated broadening.

Figure 2: XRD pattern (Run 111) with cumulative peak fits from individual pseudo-Voigt components (FCC in blue, HCP in green). SF simulation with faulting probability $\alpha = 0.3$ (blue line) is shown for comparison.

Phase	Reflection (hkl)	$\text{FWHM}_Q (\text{\AA}^{-1})$	η
HCP	100	0.12999	0.5
FCC	111	0.06629	1.0
HCP	002	0.26538	0.5
HCP	101	0.42035	0.5
FCC	200	0.1305	1.0

Table 1: Individual pseudo-Voigt fits in Q -space for representative reflections. The pseudo-Voigt shape parameter η weights Lorentz vs Gauss ($\eta = 1$ Lorentz, $\eta = 0$ Gauss).

Reviewer Comment 2.6 — A final thing I noticed was significant mismatch between observed and calculated peak positions for the FCC 220/HCP 110 reflection. This is clearly seen in the figure above where the calculated position is well to the left of the observed reflection (which is also much broader than the calculation). Since crystallites contributing this reflection have a different orientation relative to the load axis (and x-ray beam), could this be the result of anisotropic compressibility of the lattice? It seems like this should at least be commented on. In summary, it is impressive that it is possible to extract data of this quality from such extreme conditions. As intimated above, I think there may be quite a lot of complexity in the local structure and it seems as though more sophisticated models may be needed to accurately describe the diffraction features.

Reply: We agree with the referee that a noticeable misfit is observed for the FCC(220)/HCP(110) reflection. Although this discrepancy is visible in Fig. 3a & 4b, its direction and magnitude vary between runs (e.g. Run 111 vs Run 250), indicating that it is not a systematic effect. We do not believe that anisotropic compressibility is the main cause of this misfit. If lattice anisotropy were responsible, one would expect a consistent offset of the FCC(220)/HCP(110) reflection in the same direction across all runs, which is not observed (e.g. Run 111 vs Run 250 show opposite trends). Our mixed-stacking machine-learning-potential (MLP) simulations reproduce the FCC(220)/HCP(110) positions much more accurately, yielding lattice spacings in closer agreement with experiment. This suggests that the residual discrepancies in the simplified refinements reflect limitations of the refinement parameterization, which does not fully capture the microstructural complexity of the system. As already discussed in the Supplementary and emphasized by the referee, more sophisticated models will likely be needed to describe the diffraction features in detail.

Figure 3: Caption

Figure 4: Caption

Response to Reviewers : manuscript NCOMMS-25-52819-T

We would like to begin by sincerely thanking the referees for their thoughtful comments and detailed suggestions, which have been very helpful in further refining our diffraction analysis. We are also grateful for their positive assessment of our study. Acquiring diffraction data of this quality under shock-compression conditions is far from routine, and it is especially encouraging that expert referees have recognized the significance of our results. Their observations that “*it is impressive that it is possible to extract data of this quality from such extreme conditions*” and that this has “*for the first time, allowed a truly crystallographic discussion of superionic ice*” are both motivating and validating for our work.

In the following, we provide detailed point-by-point responses to all referees’ comments and questions (our responses appear in blue). For clarity, all significant changes made in the revised manuscript are also highlighted in blue.

Response to the reviewers

Reviewer 1

The target of this study, superionic ice, has recently been the subject of intense research competition, owing to its dual nature—possessing characteristics of both solids and liquids—as well as its significance in planetary science. The notable strength of this work lies in the use of XFEL and VISAR, which has enabled an unprecedentedly precise determination of temperature and pressure conditions, along with the acquisition of high-quality X-ray diffraction data. Remarkably, under such extreme high-temperature and high-pressure conditions, diffraction patterns are typically “spotty,” yet in this study, the authors obtained homogeneous and ideal “smooth” powder patterns. This has been crucial for the high-precision analysis of ice with stacking disorder, and for the first time, has allowed a truly crystallographic discussion of superionic ice. I am not a specialist in dynamic compression, and thus I am not in a position to evaluate the validity of the techniques employed in this work, such as VISAR. If the temperature and pressure in this study are indeed determined with higher accuracy and reliability than previous estimates, I have no doubt that the paper will eventually merit publication in Nature Communications. I therefore leave the judgment on this matter to other reviewers with expertise in dynamic compression. That said, there remain several issues of concern that should be carefully addressed prior to publication.

Reviewer Comment 1.1 — Interpretation of the phase diagram The primary concern lies in the authors’ claim that FCC and HCP (BCC) phases coexist as stable phases in the phase diagram. From the viewpoint of classical thermodynamics, as is evident from the Gibbs phase rule ($F - C + P = 2$), for a single-component system ($C = 1$), coexistence of two phases ($P = 2$) leaves one degree of freedom ($F = 1$), meaning that such coexistence is typically allowed only along a phase boundary line in the phase diagram. However, in Fig. 5, a certain P–T region is depicted as a coexistence region of bcc and fcc phases.

Indeed, as the authors argue, under such high-temperature conditions the free energy difference between bcc and fcc phases may be extremely small, and non-equilibrium coexistence of the two phases may occasionally occur. Nevertheless, only thermodynamically stable phases can have their

own stability fields in a phase diagram, and it is inconsistent with classical thermodynamics for two phases of different symmetry to both be stable within the same P-T region.

The only possible exception (or out of the classical thermodynamic) would be if the bcc and fcc structures were intermixed on a scale smaller than 100 nm, with no long-range periodicity maintained—essentially a nanostructured mixture. Such a nanoscale “phase separation” would not constitute a macroscopically homogeneous thermodynamic phase, and should perhaps be regarded differently in the context of a phase diagram. How to treat such nanoscopic phase-separated states in phase diagrams may be an open question for future condensed matter physics.

A similar argument applies to the mixed-stacking SI phase. At such high temperatures, it is entirely plausible that stacking disorder could be energetically more favorable than either pure fcc or hcp phases, due to the entropic gain. Stacking disorder, by its nature, lacks long-range periodicity along a given axis and therefore has no macroscopic symmetry; changes in the stacking probability α do not alter the overall symmetry. Consequently, one might interpret it as a single homogeneous phase. Indeed, in the case of ice Isd, Lupi et al. (2018, *Nature* 551, 241–245; doi:10.1038/nature24279) have pointed out that ice Isd could be more stable than either pure Ih or Ic under certain temperatures and domain sizes. In light of the above, the discussion of Fig. 5 and the purported coexistence of bcc and fcc phases should be reconsidered.

Reply:

We are very grateful to the Reviewer for raising this important point.

In our shock experiments, we observe coexistence of the bcc and fcc structures in a pressure/temperature regime, where, according to theoretical predictions, the free-energy difference between the two phases becomes extremely small. We therefore interpret this domain not as a stability field in the thermodynamic sense, but as a region of metastable or non-equilibrium coexistence reflecting the structures experimentally detected. Metastability appears to be a common feature of high-pressure ice: in static compression studies, for example, the fcc phase is known to persist metastably over the bcc phase at room temperature, despite not being the lowest free-energy phase (**Forestier et al. PRL. 2025**)

We fully agree with the Reviewer that, in light of this discussion, Figure 5 should not be regarded as a phase diagram in the strict thermodynamic sense, since we cannot determine the equilibrium stability of the indicated phases. Rather, it should be viewed as a P–T diagram (experimental observation map) showing the structures identified in our measurements. This approach is standard in high-pressure research, including studies using static compression in diamond anvil cells.

To clarify this point, we have revised both the caption of the figure and the main text to explicitly state that Figure 5 represents a P-T diagram (experimental observation map), rather than a thermodynamic equilibrium phase diagram, and that the “region of ”BCC + FCC” corresponds to experimentally observed, likely metastable, coexistence.

Reviewer Comment 1.2 — The manuscript discusses possible implications for “giant impacts”. Could the authors support this statement with examples—preferably from the metallurgy literature—where stacking disorder affects mechanical properties? Relevant citations would be welcome. Furthermore, given that ice produced under shock compression is likely to revert rapidly to ambient-pressure phases, do its mechanical or transport properties still hold planetary significance during its brief lifetime? Overly speculative implications may risk diminishing the paper’s value and should be carefully considered.

Reply: As emphasized on p. 8 of the manuscript, our experiments clearly reveal the presence of stacking disorder. However, the available data do not allow us to determine whether this disorder is

an intrinsic feature of the superionic phase or instead arises from the strong uniaxial, high-strain-rate compression associated with shock loading. Both scenarios are interesting from a fundamental point of view and potentially relevant for planetary science. If the disorder is intrinsic to superionic ice, its consequences for giant planets would be directly linked to the physical properties of the faulted ice phase. In metallurgy, stacking disorder is well known to influence strength and transport properties of metals and alloys (e.g., Su et al., *Mater. Sci. Eng. A* 803, 140696 (2021)). In planetary science, it is well known that crystalline defects play a central role in controlling rheological properties (e.g., Karato et al., *Prog. Earth Planet. Sci.* 12, 27 (2025); Sumita & Bergman, in *Treatise on Geophysics*, Elsevier, 2007) and simulations at extreme conditions are striving to include them (e.g., Li & Scandolo, *Phys. Rev. B* 109, 184108 (2024)). Stacking disorder is also known to affect viscosity and thermal conductivity, including in ice at low pressures (e.g., F. Matusalem et al, *PNAS* 119, e2203397119 (2022), Johari & Andersson, *J. Chem. Phys.* 143, 054505 (2015)). If intrinsic, such effects could therefore directly impact models of planetary interiors, transport properties, and even dynamo generation. Alternatively, if the disorder is shock-induced, it could provide important insight into the physics of planetary collisions, which cannot be reproduced by static compression alone. For instance, shocked feldspars or silicates in meteorites form maskelynite or diaplectic glass (e.g., Rubin, *Icarus* 257, 221–229 (2015)), while static experiments yield only crystalline phases. Similarly, iron alloys form equilibrium crystalline structures under static high-pressure experiments but can adopt metastable phases under shock (e.g., Boettger & Wallace, *Phys. Rev. B* 55, 5 (1997)). By analogy, shock-induced stacking disorder in superionic ice could represent a transient but relevant state during giant impacts. Whether such a metastable phase persists long enough to influence impact dynamics depends on its lifetime relative to hydrodynamic timescales, a point our current experiments cannot resolve. Given the increasing recognition of giant impacts as a key hypothesis for shaping planetary interiors (e.g., Helled et al., *Space Sci. Rev.* 216, 38 (2020); Liu et al., *Nature* 572, 355–357 (2019); Müller et al., *A&A* 638, A124 (2020); Esteves et al., *Icarus* 429, 116428 (2025); Kegerreis et al., *Astrophys. J.* 861, 52 (2018); Reinhardt et al., *MNRAS* 492, 5336–5353 (2020)), we believe it important to highlight this possibility. We note that longer-duration shock studies at these extreme conditions could be challenging, so that theoretical work and numerical simulations will be essential to address this aspect.

We agree with the referee that this aspect was not adequately discussed in the first version of the manuscript, and we have now changed the sentence at old line 197:

The identified stacking faults may significantly influence the mechanical and transport properties of superionic water [8], particularly under the extreme conditions associated with giant impacts. As a result, our understanding of ice-rich planetary interiors—such as those expected in Uranus and Neptune—could be substantially revised, given their dependence on properties like viscosity and thermal conductivity.

with

*If intrinsic, the observed stacking disorder would affect the transport properties that govern internal structure, rheology and magnetic field generation in ice-rich planets [8, Johari & Andersson, *J. Chem. Phys.* 143, 054505 (2015)]. If instead shock-induced, its significance would depend on the lifetime of the metastable faulted phase, a question that our work cannot resolve, but that future investigations should address, given the growing recognition of giant impacts as a key hypothesis for shaping planetary interiors (e.g., Helled et al., *Space Sci. Rev.* 216, 38 (2020); Liu et al., *Nature* 572, 355–357 (2019)).*

Reviewer Comment 1.3 — In Fig. S5, the CeO₂ profile does not fit well. Both the high-Q and low-Q regions appear to deviate in peak positions (with asymmetric residuals at high-Q). This may be improved by optimizing the peak shift parameters and the profile function. Although fixing $U = 0$ seems reasonable given the limited 2θ range, V should likely be treated as a variable. Also, in the caption to Fig. S5, “CeCO₂” should be corrected to “CeO₂,” and X and Y may not be asymmetric parameters but Lorentzian terms; please confirm.

Reply: We thank the Reviewer for pointing this out. We have repeated the refinement of the CeO₂ calibrant profile, this time keeping U fixed at 0 but allowing V and W to vary, together with the Zero 2θ shift. The rest of the refinement procedure remained unchanged.

This adjustment significantly improved the agreement between observed and calculated profiles across the entire Q -range, including both low- Q and high- Q regions. The final refined profile parameters are: $U = 0.0$, $V = 0.012011$, $W = 0.004258$, $X = 0.058403$, $Y = 0.057187$, $Z_{\text{zero}} = -0.0026$

The fit is now much better than in the previous version, and the asymmetric residuals at high- Q are largely suppressed. We also corrected the caption typo (“CeCO₂” \rightarrow “CeO₂”) as suggested. Finally, we confirm that in the Thompson–Cox–Hastings pseudo-Voigt function used in FullProf, X and Y are Lorentzian terms (size and strain broadening, respectively), not asymmetry parameters. This new calibration does not affect the main findings of our work. The main changes concern lattice parameters, grain sizes and micro strains, but such adjustments lie within the error bars.

Figure 1: Refinement of the CeO₂ calibrant profile using the Thompson–Cox–Hastings function. Parameters refined were V , W , X , Y , and the Zero 2θ shift, with U fixed to 0. The resulting values are $V = 0.0120$, $W = 0.0043$, $X = 0.058$, $Y = 0.057$, $Z_{\text{zero}} = -0.0026$. The fit quality is significantly improved compared to the previous versio

Reviewer Comment 1.4 — Throughout the manuscript, reflection indices, hkl , should not be enclosed in parentheses (see the introduction to International Tables for Crystallography, Vol. A).

Reply: In the revised manuscript, all reflection indices hkl have been reformatted to follow the convention of the International Tables for Crystallography, Vol. A, i.e., without parentheses.

Reviewer Comment 1.5 — The terms “stacking fault(s)” and “stacking disorder” should be used distinctly. “Stacking fault(s)” refers to individual defect(s) on one or plural layer(s) with differently

aligned to surrounding layers, whereas “stacking disorder” refers to the state of a material containing many such faults, may be defined by detectable using diffraction methods . In most cases in this manuscript, “stacking disorder” would be the more accurate term rather than “stacking fault(s)”

Reply: We thank the referee for this important remark. In the revised version of the manuscript, we now distinguish carefully between stacking fault(s), which refer to local defects in the stacking sequence, and stacking disorder, which describes the overall disordered state of the structure as observed in diffraction. Since our experimental data reveal mainly average structural effects rather than isolated planar defects, we agree that ‘stacking disorder’ is the most appropriate term in most contexts. We have therefore systematically revised the manuscript to adopt this terminology.

Reviewer Comment 1.6 — P-T conditions may be included in the legend in respective figures as well as run numbers

Reply: In the main text figures, the P–T conditions and run numbers are already included. We have updated the supplementary figures accordingly, to ensure that the P–T conditions and run numbers are systematically visible.

Reviewer Comment 1.7 — Several figure references appear as “Fig. ??.” Please check and correct them

Reply: The occurrences of “**Fig. ??**” were due to missing or broken cross-references in the manuscript. We have checked carefully all figure references and correct them to ensure consistency throughout the text.

Reviewer Comment 1.8 — On page 2 of the Supplementary Materials, “ice VII/VII*/X” is written. What does VII* mean? (Is it a typographical error for VII' ?) Since the manuscript does not explain VII', a definition should be provided

Reply: The notation VII* was indeed a typographical error, and it referred to ice VII', as defined in the articles of Hernandez and Caracas, Phys. Rev. Lett. (2016) and Queyroux et al. Phys. Rev. Lett. (2020). These papers are now cited as a reference for such phase.

Reviewer 2

This paper describes a series of x-ray diffraction measurements of ice subjected to extremes of temperature and pressure via shock compression techniques. The approach is to confine the samples using diamond windows and then subject them to a shock wave triggered by a 10ns laser pulse with a very high energy density. This creates a rather complex environment described as “elastic and plastic waves [reverberating] between the diamonds” . While the sample responds to this, it is probed using a single 50fs pulse from a free electron laser (generated by either the LCLS or XFEL facilities in two separate experiments). Thus, it effectively takes a rapid snapshot, capturing the instantaneous state of the sample. By controlling the exact delay between shock and x-ray, the experimental team thus aim to select the P/T conditions of the measurement (although determining these relies on some ancillary hydrodynamic simulations).

Reviewer Comment 2.1 — One concern I had in interpreting the diffraction data is that they correspond to signal from a gauge volume that is has a 20-30um diameter and thickness equal to that of the sample. Correspondingly, although radial gradients are minimized by the small beam diameter, any axial gradient will be averaged. This seems like an important point, but I did not see this explicitly addressed in the manuscript. If I correctly read the right-hand panel of Fig 2, it looks like there is measurable step in density around 2/3 of the way through the sample of around 5%. Which, if I look at the equations of state in Fig S17, would map on to a 10-15 GPa pressure differential. The authors state that “temperature and pressure gradients seem unlikely in [their] experiments” , but then in the SI it is mentioned that “where the sample thickness was significant, a contribution from the liquid phase was observed” . This surely indicates that there are significant gradients within the sample if liquid and solid phases co-exist. How do the authors reconcile this?

Reply: We thank the referee for raising this important point. Indeed, as correctly noted, the small X-ray beam diameter minimizes radial gradients, while the averaging across the sample thickness may include contributions from any axial gradients. This issue is discussed in Section 3 of the Supplementary Material, but we are happy to provide further clarification here. The presence and importance of axial gradients depend strongly on the hydrodynamic history of the shock compression and the probing time. To reach a representative description of these effects, we combined hydrodynamic simulations with optical diagnostics (see Section 2 of the Supplementary Material, Fig. 2 and Fig. S1). As the referee observes, Fig. 2 does display a density “step”. This feature arises in the LCLS data because reverberations from the 30 μm diamond ablator overtake the first shock waves within the water layer (Fig.S1 a,b,c). As a result, a steep gradient forms, with a colder region produced by more numerous reverberations, and a hotter region resulting from fewer compressions. In this case, two sets of distinct thermodynamic conditions are generated (Fig. S4a,b). Although both regions contribute to the diffraction patterns, we find that the hotter part is close to (or above) melting (Fig. S4a,b and Fig. S18) and primarily adds a liquid background signal. By contrast, the crystalline diffraction signal originates predominantly from the colder region. This interpretation is consistent with the need to include a substantial liquid background component when refining the LCLS diffraction data (Section 4 of the Supplementary Material). In contrast, for the EuXFEL experiments, we used thinner water layers ($\sim 25\text{--}30 \mu\text{m}$), which prevented the overtaking of shock reverberations and produced much more uniform conditions (Fig. S1d and Fig. S4c,d). The resulting diffraction data are consistent with those from the “cold” part of the LCLS experiments and show similar features under comparable conditions (see Fig. S8 and Fig. S9). This provides further support for our interpretation.

Finally, regarding the quoted sentence “temperature and pressure gradients seem unlikely in [their] experiments”: we agree that, when read in isolation, this wording may appear ambiguous. In its context (line 169 of the old version of the manuscript), the full sentence reads: “However, since the occurrence of similar temperature and pressure gradients seems unlikely in our experiments, this finding raises the possibility of a genuine coexistence of BCC-SI and FCC-SI phases, prompting further discussion on the underlying mechanisms and kinetics.” This sentence was specifically included to address the simultaneous observation of BCC and FCC structures in both static and dynamic experiments. Our intention was not to deny the presence of gradients in the shock-compression data, but rather to emphasize that such gradients are unlikely to match those in static experiments.

Reviewer Comment 2.2 — Of course, the data themselves contain information on any strain gradient. The authors helpfully included data and refinement details from a CeO₂ standard, so the instrument resolution can be accounted for. With this information, substantial sample broadening

is clearly present in all the datasets shown in Fig 3, although it is difficult to quantitatively assess this just by looking at the figures and any strain could also be convolved with particle size effects.

Reply: The broadening of the lines associated to the grain size effects has been evaluated and discussed in the supplementary material, section 6.5. We find that the grain size broadening has a negligible effect on the widths of the collected diffraction peaks.

Reviewer Comment 2.3 — The diffraction profiles observed in the higher P/T region of the phase diagram do convincingly show both FCC and HCP features. And it seems reasonable to interpret these in terms of stacking faults. But I have some concerns regarding the quality of the Le Bail fits. All the patterns in Fig 3 seem to contain Bragg features that are much sharper than simulated peaks. These are easily discernable in the difference curves, circled in the image below:

These sharp features appear more-or-less exactly at the expected positions for the respective crystalline phases, so could be explained by a handful of fault-free crystallites, or extended fault-free domains within crystallites with correspondingly sharper peaks. Having a distribution of domain sizes seems to be rather likely.

Reply: We appreciate the referee's detailed comments and for highlighting the differences observed in the curves of Fig. 3. We agree that it is indeed plausible that our data reflect a distribution of domain sizes. However, we believe it is difficult to infer the origin of such a distribution solely from the available data. As the referee points out, similarly sharp features are also visible in the BCC phase (Fig. 3c), where the data can be explained without invoking faults. We therefore consider it more likely that these features arise from the nucleation process associated with compression. Experimentally, sharper features than those obtained from calculated refinements have also been reported for shocked SiO_2 (see Supplementary Fig. 2. of **A.E. Gleason et.al., Nat. Com. 2015**), which may support our interpretation. Nevertheless, we fully agree with the referee that future studies should examine these features in greater detail.

Reviewer Comment 2.4 — There are also significant discrepancies between fit and data in the SF refinements. This is most notable in Fig 4b below. There are clearly “lumps” in the data that are not fitted by the model in the region from 2.7 to 3.5 \AA^{-1} . I think this is probably indicating that the peak profile model is not adequate to capture the complexities of the real system. It seems plausible that different directions in the lattice could retain coherence over longer distances than others.

Reply: We fully agree with the referee that the local structure may exhibit significant complexity that our simple intrinsic stacking model does not capture. As we explicitly state in Section 5 of the Supplementary Material, “the actual microstructure is likely more complex than a simple FCC–HCP layer alternation.” Nonetheless, we chose not to overinterpret our data by attempting to construct a more complex structural model.

Reviewer Comment 2.5 — Usually, the profile parameters in refinement are parameterized in a way that would not capture hkl-dependent effects on peak broadening. I wondered if simple individual peak fits might describe the data better. And these might contain interesting information, for example, I was curious if either the HCP or FCC component had measurably and consistently sharper peaks than the other.

Reply: Thank you for this helpful suggestion. We performed individual pseudo-Voigt fits in Q -space for representative reflections (HCP: 100, 002, 101; FCC: 111, 200), as summarized in Table 1. On average, the HCP reflections are broader (mean $\text{FWHM}_Q \sim 0.27 \text{ \AA}^{-1}$) compared to the FCC reflections (mean $\text{FWHM}_Q \sim 0.10 \text{ \AA}^{-1}$). However, the broadening within HCP is not uniform: HCP(100) is comparably sharp to FCC(111), whereas HCP(002) and HCP(101) are substantially broader. We also observed that the HCP peaks tend to have a shape factor $\eta \approx 0.5$, indicating a more Gaussian-like profile, whereas the FCC peaks are more Lorentzian ($\eta \approx 1$). These results are consistent with what we already presented in the Supplementary Materilas (see SM section 6.2), where separate FCC and HCP fits revealed similar trends, in line with disorder-dominated broadening.

Figure 2: XRD pattern (Run 111) with cumulative peak fits from individual pseudo-Voigt components (FCC in blue, HCP in green). SF simulation with faulting probability $\alpha = 0.3$ (blue line) is shown for comparison.

Phase	Reflection (hkl)	$\text{FWHM}_Q (\text{\AA}^{-1})$	η
HCP	100	0.12999	0.5
FCC	111	0.06629	1.0
HCP	002	0.26538	0.5
HCP	101	0.42035	0.5
FCC	200	0.1305	1.0

Table 1: Individual pseudo-Voigt fits in Q -space for representative reflections. The pseudo-Voigt shape parameter η weights Lorentz vs Gauss ($\eta = 1$ Lorentz, $\eta = 0$ Gauss).

Reviewer Comment 2.6 — A final thing I noticed was significant mismatch between observed and calculated peak positions for the FCC 220/HCP 110 reflection. This is clearly seen in the figure above where the calculated position is well to the left of the observed reflection (which is also much broader than the calculation). Since crystallites contributing this reflection have a different orientation relative to the load axis (and x-ray beam), could this be the result of anisotropic compressibility of the lattice? It seems like this should at least be commented on. In summary, it is impressive that it is possible to extract data of this quality from such extreme conditions. As intimated above, I think there may be quite a lot of complexity in the local structure and it seems as though more sophisticated models may be needed to accurately describe the diffraction features.

Reply: We agree with the referee that a noticeable misfit is observed for the FCC(220)/HCP(110) reflection. Although this discrepancy is visible in Fig. 3a & 4b, its direction and magnitude vary between runs (e.g. Run 111 vs Run 250), indicating that it is not a systematic effect. We do not believe that anisotropic compressibility is the main cause of this misfit. If lattice anisotropy were responsible, one would expect a consistent offset of the FCC(220)/HCP(110) reflection in the same direction across all runs, which is not observed (e.g. Run 111 vs Run 250 show opposite trends). Our mixed-stacking machine-learning-potential (MLP) simulations reproduce the FCC(220)/HCP(110) positions much more accurately, yielding lattice spacings in closer agreement with experiment. This suggests that the residual discrepancies in the simplified refinements reflect limitations of the refinement parameterization, which does not fully capture the microstructural complexity of the system. As already discussed in the Supplementary and emphasized by the referee, more sophisticated models will likely be needed to describe the diffraction features in detail.

Figure 3: Caption

Figure 4: Caption

Review Andriambarijaona et al

This paper describes a series of x-ray diffraction measurements of ice subjected to extremes of temperature and pressure via shock compression techniques. The approach is to confine the samples using diamond windows and then subject them to a shock wave triggered by a 10ns laser pulse with a very high energy density. This creates a rather complex environment described as “elastic and plastic waves [reverberating] between the diamonds”. While the sample responds to this, it is probed using a single 50fs pulse from a free electron laser (generated by either the LCLS or XFEL facilities in two separate experiments). Thus, it effectively takes a rapid snapshot, capturing the instantaneous state of the sample. By controlling the exact delay between shock and x-ray, the experimental team thus aim to select the P/T conditions of the measurement (although determining these relies on some ancillary hydrodynamic simulations).

One concern I had in interpreting the diffraction data is that they correspond to signal from a gauge volume that is has a 20-30um diameter and thickness equal to that of the sample. Correspondingly, although radial gradients are minimized by the small beam diameter, any axial gradient will be averaged. This seems like an important point, but I did not see this explicitly addressed in the manuscript. If I correctly read the right-hand panel of Fig 2, it looks like there is measurable step in density around 2/3 of the way through the sample of around 5%. Which, if I look at the equations of state in Fig S17, would map on to a 10-15 GPa pressure differential. The authors state that “temperature and pressure gradients seem unlikely in [their] experiments”, but then in the SI it is mentioned that “where the sample thickness was significant, a contribution from the liquid phase was observed”. This surely indicates that there *are* significant gradients within the sample if liquid and solid phases co-exist. How do the authors reconcile this?

Of course, the data themselves contain information on any strain gradient. The authors helpfully included data and refinement details from a CeO₂ standard, so the instrument resolution can be accounted for. With this information, substantial sample broadening is clearly present in all the datasets shown in Fig 3, although it is difficult to quantitatively assess this just by looking at the figures and any strain could also be convolved with particle size effects.

The diffraction profiles observed in the higher P/T region of the phase diagram do convincingly show both FCC and HCP features. And it seems reasonable to interpret these in terms of stacking faults. But I have some concerns regarding the quality of the Le Bail fits. All the patterns in Fig 3 seem to contain Bragg features that are much sharper than simulated peaks. These are easily discernable in the difference curves, circled in the image below:

These sharp features appear more-or-less exactly at the expected positions for the respective crystalline phases, so could be explained by a handful of fault-free crystallites, or extended fault-free domains within crystallites with correspondingly sharper peaks. Having a distribution of domain sizes seems to be rather likely.

There are also significant discrepancies between fit and data in the SF refinements. This is most notable in Fig 4b below

There are clearly “lumps” in the data that are not fitted by the model in the region from 2.7 to 3.5 \AA^{-1} . I think this is probably indicating that the peak profile model is not adequate to capture the complexities of the real system. It seems plausible that different directions in the lattice could retain coherence over longer distances than others. Usually, the profile parameters in refinement are parameterized in a way that would not capture hkl -dependent effects on peak broadening. I wondered if simple individual peak fits might describe the data better. And these might contain interesting information, for example, I was curious if either the HCP or FCC component had measurably and consistently sharper peaks than the other.

A final thing I noticed was significant mismatch between observed and calculated peak positions for the FCC 220/HCP 110 reflection. This is clearly seen in the figure above where the calculated position is well to the left of the observed reflection (which is also much broader than the calculation). Since crystallites contributing this reflection have a different orientation relative to the load axis (and x-ray beam), could this be the result of anisotropic compressibility of the lattice? It seems like this should at least be commented on.

In summary, it is impressive that it is possible to extract data of this quality from such extreme conditions. As intimated above, I think there may be quite a lot of complexity in the local structure and it seems as though more sophisticated models may be needed to accurately describe the diffraction features.